# Influences of CO<sub>2</sub> and Fungus-Assisted Bioweathering

| 2           | on Fluoridated Apatite                                                                                                                                                                                                                                                                                                                                                                             |
|-------------|----------------------------------------------------------------------------------------------------------------------------------------------------------------------------------------------------------------------------------------------------------------------------------------------------------------------------------------------------------------------------------------------------|
| 3<br>4<br>5 | Mu Su <sup>1,2#</sup> , Yuyang Cao <sup>2#</sup> , Lingyi Tang <sup>2#</sup> , Haoyang Weng <sup>2#</sup> , Xiaoqing Shao <sup>2</sup> , Zhijun Wang <sup>2</sup> , Gilberto de Oliveira Mendes <sup>3</sup> , Lingzi Meng <sup>2</sup> , Meiyue Xu <sup>2</sup> , Kun He <sup>4*</sup> , Zibo Li <sup>1</sup> , Mao Luo <sup>1</sup> , Geoffrey Michael Gadd <sup>5</sup> , Zhen Li <sup>2*</sup> |
| 6<br>7<br>8 | <sup>1</sup> State Key Laboratory of Palaeobiology and Stratigraphy, Nanjing Institute of Geology and Palaeontology, Chinese Academy of Sciences, Nanjing 210008, China                                                                                                                                                                                                                            |
| 9<br>10     | <sup>2</sup> College of Resources and Environmental Sciences, Nanjing Agricultural University, Nanjing,<br>Jiangsu 210095, China                                                                                                                                                                                                                                                                   |
| 11<br>12    | <sup>3</sup> Laboratório de Microbiologia e Fitopatologia, Instituto de Ciências Agrárias, Universidade<br>Federal de Uberlândia, Monte Carmelo, MG, Brazil                                                                                                                                                                                                                                        |
| 13<br>14    | <sup>4</sup> Key Laboratory of Petroleum Geochemistry, Research Institute of Petroleum Exploration and Development, China National Petroleum Corporation, Beijing 100083, China                                                                                                                                                                                                                    |
| 15<br>16    | <sup>5</sup> Geomicrobiology Group, School of Life Sciences, University of Dundee, Dundee DD1 5EH,<br>Scotland, UK                                                                                                                                                                                                                                                                                 |
| 17          | *These authors contributed equally to this work                                                                                                                                                                                                                                                                                                                                                    |
| 18          | Corresponding author: Zhen Li (lizhen@njau.edu.cn)                                                                                                                                                                                                                                                                                                                                                 |
| 19          |                                                                                                                                                                                                                                                                                                                                                                                                    |
| 20          |                                                                                                                                                                                                                                                                                                                                                                                                    |
| 21          |                                                                                                                                                                                                                                                                                                                                                                                                    |
| 22          |                                                                                                                                                                                                                                                                                                                                                                                                    |
| 23          |                                                                                                                                                                                                                                                                                                                                                                                                    |
| 24          |                                                                                                                                                                                                                                                                                                                                                                                                    |
| 25          |                                                                                                                                                                                                                                                                                                                                                                                                    |
| 26          |                                                                                                                                                                                                                                                                                                                                                                                                    |
| 27          |                                                                                                                                                                                                                                                                                                                                                                                                    |

#### Abstract

Fluoridated apatite (FAP) is the dominant P source for ecosystems. However, the bioweathering of FAP is still not fully elucidated. In this study, phosphate-solubilizing fungus (Aspergillus niger) was firstly incubated in soil to examine the weathering of FAP, on both cross and longitudinal sections. It showed that the fungus induced more pronounced erosion channels on the cross sections in a P-deficient soil. We therefore aimed to disentangle the relative contributions of biological (phosphate-solubilizing fungus) and abiotic factors (CO2 and crystal face orientation) to the observed weathering contrasts. To further investigate the weathering contrasts on different sections of FAP, incubation was conducted in a culture medium. Fungal colonization on the cross section of FAP resulted in deeper P depletion zones and enriched secondary minerals (primarily calcium oxalate) than those on the longitudinal section. Additionally, elevated CO2 (10%) significantly accelerated the weathering of FAP on the cross section, which was confirmed by its enhanced surface roughness, further promoted fungal colonization and subsequent bioweathering for FAP. Synergistic interactions between fungi and elevated CO2 accelerate phosphate mineral weathering, providing a new insight on P cycling in soil microenvironments such as the rhizosphere.

Key words: Fungal bioweathering; Apatite; Elevated CO<sub>2</sub>; Cross section; Hyphal-mineral interface





















#### 1. Introduction

Phosphorus is a key element in the replication, structural composition and metabolism of biomolecules (Elser, 2012). However, organisms are generally limited by available P in natural ecosystems (Elser et al., 2007; Meng et al., 2024). Therefore, P sets the pace of terrestrial and marine biological productivity on geological timescales (Tyrrell, 1999). FAP provides primary P for ecosystems via weathering (Chapin et al., 2011; Gadd, 2017). The delicate changes in apatite (the largest P inventory on Earth) weathering may significantly affect P availability in ecosystems on geological timescales. Fungi can accelerate weathering of phosphate minerals through a series of growth and metabolic strategies (Hoffland et al., 2004; Rawat et al., 2021). For instance, phosphatesolubilizing fungi (PSF) can dissolve phosphate minerals via secreting proton and metabolic products (i.e., organic acids and siderophores) (do Nascimento et al., 2021; Mendes et al., 2021; Rawat et al., 2021). Filamentous fungi can also accelerate mineral destruction via hyphal osmotic pressure (Bonneville et al., 2016; Bonneville et al., 2009). Complementary processes allow the filamentous PSF to substantially erode phosphate minerals (Hoffland et al., 2004; Smits et al., 2012). CO<sub>2</sub> is a crucial abiotic factor influencing FAP weathering. Dissolved CO<sub>2</sub> can induce formation of carbonic acid, which enhances the apatite dissolution by providing additional protons (Filippelli, 2008). In some natural environments, CO<sub>2</sub> concentration can be very high. For example, rhizosphere and similar microbial habitats often exhibit 10-100 times higher CO<sub>2</sub> than atmosphere from root and microbial respiration (Drever, 1994). Similarly, confined microenvironments such

https://doi.org/10.5194/egusphere-2025-5242 Preprint. Discussion started: 12 November 2025 © Author(s) 2025. CC BY 4.0 License.






















as soil pores, fungal mats, or decaying organic-rich sediments may accumulate elevated CO<sub>2</sub>, leading to locally acidified conditions. Elevated CO2 not only promotes chemical dissolution of apatite but also alters microbial activity and growth patterns (Bertoloni et al. 2006; Schulz et al. 2012). The interplay between CO<sub>2</sub>-driven geochemical reactions and CO<sub>2</sub>-mediated microbial responses may strongly regulate apatite weathering in localized niches, yet this coupled mechanism remains poorly constrained. Weathering of apatite is also regulated by mineralogical properties (Bogomolova et al., 2003). The weathering rates on different sections of a mineral could be distinct because of different bond values or hydration energies (Mkhonto and de Leeuw, 2002; Ruiz-Agudo and Putnis, 2012). Different sections of FAP crystals have distinct structures and atomic arrangements (Figs. S1a, b) (Elliott, 2002; Ichijo et al., 1992). In the FAP cross section (C<sub>FAP</sub>), the Ca, P, F are arranged as the units of hexagons (Fig. S1a), while in the FAP longitudinal section (L<sub>FAP</sub>), they form a unit of rectangle (Fig. S1b). The P densities in the C<sub>FAP</sub> and L<sub>FAP</sub> are 7.8 and 7.1 P atom exposed per nm<sup>-</sup> <sup>2</sup> (Ichijo et al., 1992). The difference in surface elemental distribution results in higher proportion of P atoms (40% of P atoms in the total number of atoms) in  $C_{FAP}$  compared to  $L_{FAP}$  (25%). In addition, orientation within mineral crystals may influence chemical stability and mechanical resistance (Giri et al., 2012). Different crystal planes of the mineral showed distinct elemental composition (Gleeson et al., 2005; Ichijo et al., 1992) and surface properties (e.g., porosity, microtopography, surface charge, hydrophobicity) (Bogomolova et al., 2003), which would influence microbial growth on mineral surfaces (Zhang et al., 2021). To maximize nutrient and minimize energy consumption, microorganisms selectively attach onto mineral surfaces

containing growth-limiting nutrients, (i.e., P, N) (Rogers et al., 1998). Cell adhesion to minerals was usually correlated with particular element site density (Neal et al., 2005). In addition, microorganisms have a strong tendency to grow along surface cracks and junctions between crystals (Bogomolova et al., 2003). However, the contrasts of fungal bioweathering on the different sections of FAP have not been elucidated.

The aim of this study was to investigate how elevated  $CO_2$  and A. niger affect FAP weathering. The effect of fungal colonization and  $CO_2$  concentration on weathering  $C_{FAP}$  and  $L_{FAP}$  was explored.

# 2. Materials and Methods

## 2.1 Fungal Strain Preparation

The Aspergillus niger (CCTCC No. M2023240) was applied in this study. A. niger was cultured on potato dextrose agar (PDA) medium at 28 °C for 5 d to produce spores. The PDA medium had 5 g/L potato infusion, 20.0 g/L glucose, and 14.0 g/L agar. The medium was drenched with sterile water and the spores were carefully scraped from the plate surface with a fine artist's brush. The suspension was then filtered through a three-layer sterile cheesecloth to eliminate mycelial fragments.

# 2.2 Preparation of FAP

Durango fluorapatite (FAP) was considered as a standard geological apatite. The FAP was sectioned into 200  $\mu$ m thick, 0.5 cm  $\times$  0.5 cm slices with cross sections (Figs. S1A and S1C) and longitudinal sections (Figs. S1B and S1C). The longitudinal section of an appetite slice was cut along the edges of the unit cell. All FAP slices were stored in a desiccator prior to the following

experiments. A three-dimensional structure of the apatite cross and longitudinal sections is shown in Fig. S1A and Fig. S1B, respectively.

# 2.3 Bioweathering of FAP in Soil

Bioweathering of FAP by *A. niger* was first conducted using P-deficient soil. The P-deficient soil sample (0–20 cm) was collected from Chongzuo, Guangxi, China. The soil was acidic (pH 4.95) and contained only 0.53 mg Kg<sup>-1</sup> of available P. Soil sample was sieved (2 mm mesh) to remove roots and gravel. Then 20 g dry soil was placed in a 50 mL glass bottle. After sterilization at 121 °C for 1 h, each bottle content was evenly sprayed with 1 mL sterile solution (containing 0.7 g of C and 0.028 g of N per week per kg dry soil). Subsequently, 1 mL *A. niger* spore suspension (4.3×10<sup>7</sup> spores mL<sup>-1</sup>) was inoculated into the soil. Afterwards, two ultraviolet-sterilized glass sheets with attached FAP samples (cross and longitudinal sections) were inserted vertically into the soil. These two treatments were denoted as CPSF@Soil and LPSF@Soil. Each treatment was performed with two experimental replicates. The bottles were incubated in the dark at 25 °C (60% of maximum soil water-holding capacity). After 30 d incubation, two FAP slices were cleaned using sterile water and air dried at room temperature for SEM analysis.

#### 2.4 Weathering of FAP in Medium

(i) Weathering of FAP by PSF: To further investigate the factors causing differences in the weathering rates of different apatite crystal facets, a set of experiments was conducted in PDA medium. Two ultraviolet-sterilized glass sheets with attached FAP samples were vertically inserted into the PDA medium. Then, 0.1 mL of *A. niger* spore suspension (4.3×10<sup>7</sup> spores mL<sup>-1</sup>) was inoculated. The FAP slices with *A. niger* inoculation were incubated at 28 °C under sterile ambient

air for 45 d, which were denoted as CPSF (cross section with *A. niger*) and LPSF (longitudinal section with *A. niger*) treatments (Fig. S1D). Each treatment was performed with two experimental replicates. Two FAP slices were then cleaned using sterile water and air dried at room temperature for SEM-EDS, Raman imaging and scanning electron microscopy (RISE) analysis. In addition, FAP with growing hyphae were protected by deposition of a platinum strap (~1 mm thick). Then, focused ion beam (FIB) was applied to prepare hypha-mineral interface profiles (~90 nm thick) for transmission electron microscopy-energy dispersive spectroscopy (TEM-EDS) analysis.

- (ii) Weathering of FAP by elevated CO<sub>2</sub>: FAP slices were also prepared without PSF addition and incubated at 10% CO<sub>2</sub> concentration (denoted as C<sub>FAP</sub>@CO<sub>2</sub> and L<sub>FAP</sub>@CO<sub>2</sub> treatments) to investigate the effects of sole CO<sub>2</sub> on apatite weathering. Each treatment was performed with two experimental replicates. After 45 d incubation, FAP slices were gently rinsed using sterile water and air dried at room temperature for scanning electron microscopy and atomic force microscopy (AFM).
- (iii) Weathering of FAP by PSF and elevated CO<sub>2</sub>: To examine the synergistic effect of fungal colonization and elevated CO<sub>2</sub> on apatite weathering, another two glass sheets attached FAP samples with PSF addition were incubated in an incubator containing 10% CO<sub>2</sub> (denoted as CPSF@CO<sub>2</sub> and LPSF@CO<sub>2</sub> treatments). Each treatment was performed with two experimental replicates. After 45 d incubation, two FAP slices were cleaned using sterile water and air dried at room temperature for SEM analysis.

# 2.5 Instrumentation

SEM image analysis was performed using a Carl Zeiss Supra 55 system (Carl Zeiss Inc.,



















Germany) in secondary electron imaging mode with an acceleration voltage of 7-15 kV. To enhance image quality and minimize charging, samples were sputter-coated with gold for SEM analysis. To ensure representation, the regions scanned were selected randomly. Semi-quantitative analysis (collecting time: 60 s) was performed using an Oxford Aztec X-Max 150 energy dispersive spectrometer (EDS). RISE system consisted of a Wissen schaftliche Instrumente and Technologie GmbH (WITech, Germany) Alpha 300 confocal Raman microscope combined with an SEM (TESCAN-VEGA3). The spectral region of 0-4500 cm<sup>-1</sup> was recorded using a 532 nm laser (15 mW with 4×15 s scans). In addition, Raman spectra of P-O (for PO<sub>4</sub><sup>3-</sup>in apatite slice) vibration were collected using a Horiba Lab RAM HR Evolution instrument equipped with an Olympus microscope. The instrument was calibrated using a silicon wafer at band position of 520.7 cm<sup>-1</sup>. Then, the spectral region of 400–1500 cm<sup>-1</sup> was recorded using a 633 nm laser (4 mW with 2 ×60 s scans). Hypha-apatite interface profiles were prepared and polished using Zeiss Crossbeam 550 FIB-SEM. Polished samples were then analyzed using TEM-EDS (deadtime: 90 s). TEM was performed using FEI Tecnai G2 F20 system. Semi-quantitative analysis (collecting time: 40 s) was performed using an Oxford Aztec X-Max 150 EDS. 170 AFM measurement was carried out using a Bruker Dimension Icon scanning probe microscope at room temperature. The PeakForce Tapping mode was utilized for all AFM measurements. The tip (Bruker, TAP150A) had a half cone angle of 10°, a radius of 10 nm, and a resonance frequency of 150 kHz. The nominal force constant was 5 N/m and the actual spring 173 constant of the probe was calibrated before each test.

## 2.6 Data Analysis

Effects of elevated  $CO_2$  on the surface roughness of FAP cross and longitudinal sections were evaluated by comparing  $C_{FAP}@CO_2$  and  $L_{FAP}@CO_2$  treatments. Data was tested for normality and homogeneity of variances and then analyzed by a One-way analysis of variance at a significance level of P < 0.05. All statistical analyses were conducted in R software (version 4.3.2; R Core Team, 2023). The 3D structural diagram of apatite sections was examined by CrystalMaker 10.4.

#### 3. Results

# 3.1 Fungal-induced Surface Dissolution and Secondary Mineral Formation on FAP

Hyphae extended to FAP surfaces in soil experiments, with deep channels forming along hyphae in CPSF@Soil (Figs. 1a, c) while none forming in LPSF@Soil (Figs. 1b, d). For the incubation in the medium, more fungal mycelia were observed in CPSF compared with LPSF (Figs. 2a, b). Abundant secondary minerals formed around mycelium on both sections (Figs. 2c, e, d, f), with more bipyramid-shaped minerals on  $C_{FAP}$  (Fig. 2c). EDS analysis showed intense C, O, and Ca signals in these secondary minerals. Based on their morphology, these minerals resembled calcium oxalate (Mendes et al., 2022).

RISE analysis was applied to determine the phase of secondary minerals on the FAP surface.

The peak at 968 cm<sup>-1</sup> corresponded to the  $v_1$  stretching vibration of P-O from PO<sub>4</sub><sup>3-</sup> (Fig. S2). The 1488 cm<sup>-1</sup> peak was assigned to v (C-O) vibrations of oxalate (Su et al., 2023). In the spot analysis on initial  $C_{FAP}$  and  $L_{FAP}$ , only the characteristic peaks of phosphate were observed (Fig. S2). After incubation, the characteristic peaks of oxalates were detected on the rhomboid-shaped secondary





















minerals on both sections (Fig. S2).

# 3.2 Fungal Influence on Ca and P Migration on the FAP Surface

582, 605 cm<sup>-1</sup> to  $v_4$ , the peak at 963 cm<sup>-1</sup> to  $v_1$ , and those at 1032, 1058, 1078 cm<sup>-1</sup> to  $v_3$  P-O vibrations. The peaks at 446, 1032, 1058, 1078 cm<sup>-1</sup> were only observed on the C<sub>FAP</sub> (Fig. 3a), while the characteristic peak at 429 cm<sup>-1</sup> was only observed on the L<sub>FAP</sub> (Fig. 3b). Additionally, on the C<sub>FAP</sub>, the P-O peaks (e.g., 582, 605, 1032, 1058, 1078 cm<sup>-1</sup>) exhibited higher intensities 20 μm from the hypha (point a1) than at 2 µm (point a2) (Fig. 3a; Table S1). The characteristic peaks of P-O remained consistent on the  $L_{FAP}$  (Fig. 3b). FIB-prepared hyphal-mineral interface profiles detected altered Ca and P atomic content (Figs. 3c, d, e). At 0.03 μm depth from the hyphal-mineral interface, Ca and P contents on C<sub>FAP</sub> were only 3.5% and 2.3% (Fig. 3c), versus 4.9% and 6.4% on L<sub>FAP</sub> (Fig. 3d). The atomic contents of Ca and P in both profiles increased rapidly from the hyphal-mineral interface to a depth of 0.1  $\mu$ m, where they reached 12.4% and 9.1% for the C<sub>FAP</sub>, 15.7% and 13.0% for the L<sub>FAP</sub>. Ca and P contents stabilized at 19.6% and 14.3% at 3.2 µm depth in C<sub>FAP</sub>, versus 20.7% and 15.7% at 1.7 μm in L<sub>FAP</sub>. In contrast to L<sub>FAP</sub>, the contents of Ca and P at the same depth in the C<sub>FAP</sub> were lower (Figs. 3c, d). EDS analysis showed that the Ca/P ratios in the C<sub>FAP</sub> profile was always higher than that in the L<sub>FAP</sub> profile within a depth of 0.32 µm from the hyphal-mineral interface. Then, the Ca/P ratios tended to be constant ( $\sim$ 1.3) in both the profiles (Fig. 3e).

Raman spectra showed the intense P-O signal on the FAP surfaces in both CPSF and LPSF

treatments (Figs. 3a, b). The peaks at 429, 446 cm<sup>-1</sup> were assigned to v<sub>2</sub> P-O vibrations, those at

Both  $C_{FAP}$  and  $L_{FAP}$  exhibited relatively smooth surfaces before incubation (Figs. 4a, c). Elevated  $CO_2$  exposure produced abundant sharp hexagonal columnar structures (2-3  $\mu$ m diameter) on  $C_{FAP}$  (Fig. 4b), while produced trench-like structures on  $L_{FAP}$  (Fig. 4d). Initial  $C_{FAP}$  step height varied -2 to 3 nm (Fig. 4e), expanding to -9-9 nm after elevated  $CO_2$  incubation (Fig. 4f). In contrast, the step height variation for the  $L_{FAP}$  ranged from -6 to 6 nm (Figs. 4g, h). Elevated  $CO_2$  significantly increased the surface roughness of the  $C_{FAP}$  (from 2.8 to 5.4 nm), while had limited effect on the surface of  $L_{FAP}$  (Fig. 4i).

# 3.4 Synergistic Effects of CO<sub>2</sub> and Fungi on FAP Weathering

Abundant hexagonal columnar structures were formed on the surface of C<sub>FAP</sub> in the CPSF@CO<sub>2</sub> treatment (Fig. 5a). However, the sharpness of the columnar structures was reduced compare with that in the C<sub>FAP</sub>@CO<sub>2</sub> treatments (Fig. 4b). For the L<sub>FAP</sub>, elevated CO<sub>2</sub> and PSF synergistically induced additional cracks in LPSF@CO<sub>2</sub> (Fig. 5d). Compared with L<sub>FAP</sub>, 40.2 mm<sup>2</sup> more secondary mineral areas were formed on the surface of C<sub>FAP</sub> (Fig. S3). These secondary mineral areas were mainly composed of calcium oxalate particles (Figs. 5b, e). Calcium oxalate particles predominantly accumulated along hyphae on C<sub>FAP</sub> (Fig. 5c), whereas on L<sub>FAP</sub> they exhibited scattered low-density aggregation (Fig. 5f).

# 4. Discussion

The different sections of FAP are significantly different in elemental distribution (Figs. S1a, b) and structural stability. The Ca, P, and F are arranged as the units of hexagons in the  $C_{FAP}$  (Fig. S1a), while forming a unit of rectangle in the  $L_{FAP}$ . The difference in surface elemental distribution results in a higher proportion of P atoms (40% of P atoms in the total number of atoms) in  $C_{FAP}$ 





















compared to  $L_{FAP}$  (25%). In addition, the screw dislocation of atoms often causes surface defects of crystal, which further reduces the structural stability of mineral crystals. Since the screw dislocation of atoms often occurs in the cross section of the crystal (Cuisinier et al., 1992),  $C_{FAP}$  may have lower resistance to weathering than  $L_{FAP}$ . These differences in elemental distribution and structural stability between  $C_{FAP}$  and  $L_{FAP}$  may further influence weathering of FAP.

A. niger induced more P depletion zones on the  $C_{FAP}$  (Figs. 3a, c), suggesting that  $C_{FAP}$  is more vulnerable to fungal weathering than L<sub>FAP</sub>. PSF usually grow and expand their hyphae on the surface of minerals to obtain nutrients (Gadd, 2010). The relatively high P density in C<sub>FAP</sub> may lead to the preferential growth of fungal hyphae. The low P content and high Ca/P ratio (Fig. 3e) at the mycelia-FAP interface further suggest that the P acquisition by A. niger may cause stronger biological weathering in C<sub>FAP</sub>. Fungal hyphae can accelerate the weathering of FAP via the following two pathways. First, organic acids released from the TCA cycle accelerate the solubilization of apatite mineral (Rawat et al., 2021; Su et al., 2021). Oxalic acid, the most abundant organic acid secreted by A. niger have higher acidity constants than many other organic acids (Zhang et al., 2021). Given the low structural stability of C<sub>FAP</sub>, H<sup>+</sup> and organic acid ligands drive preferential bioweathering. Oxalic acid can react with Ca2+ dissolved from FAP, the formation of calcium oxalate therefore can be applied to evaluate such fungal bioweathering processes (Sturm et al., 2015; Su et al., 2021). The results that more calcium oxalate were formed near the mycelium on the C<sub>FAP</sub> than L<sub>FAP</sub> (Figs. 2c, d), further confirmed that the bioweathering of C<sub>FAP</sub> is stronger than that of L<sub>FAP</sub>. Second, A. niger can also accelerate the physical destruction of FAP through the biomechanical forces of mycelium growth. Fungal appressoria can produce

osmotic pressures of up to 10– $20~\mu N/\mu m^2$  during hyphal growth, which would substantially accelerate the physical weathering (Hoffland et al., 2004; Howard et al. 1991). Screw dislocations in crystal cross-sections destabilize  $C_{FAP}$  structure, enhancing its susceptibility to biomechanical destruction. Moreover, the released fluorine from FAP did not cause evident toxicity on the mineral surface.

CO<sub>2</sub> can form carbonic acid in humid environments, resulting in FAP weathering. In this study, elevated CO<sub>2</sub> induced the formation of hexagonal pyramids and trench-like structures (Figs. 4a, b, c, d) on FAP surfaces and increased the roughness on C<sub>FAP</sub> surface (Figs. 4e, f, i). This suggests that abiotic factors also preferably weather C<sub>FAP</sub>. In addition, the decreased sharpness of the columnar structure on the C<sub>FAP</sub> and more cracks on the L<sub>FAP</sub> after fungal inoculation indicated that the PSF would survive and perform its solubilizing ability under elevated CO<sub>2</sub> (Figs. 5a, d). These observations highlight that elevated CO<sub>2</sub> not only accelerates chemical dissolution but also sustains fungal colonization and activity on mineral surfaces.

Elevated CO<sub>2</sub> can promote surface roughness of FAP, especially on CFAP, which provides more favorable habitats for the spores and hyphae of PSF. Such roughened surfaces may facilitate adhesion, enhance hyphal penetration, and stimulate oxalic acid secretion by PSF, further promoting bio-weathering. The synergistic effect of fungal colonization and high CO<sub>2</sub> concentration on FAP weathering has many important implications. In soils, rhizosphere serves as the crucial zone for P transfer from the lithosphere to the biosphere, which mediates crucial P transfer from lithosphere to biosphere. Rhizosphere zones often maintain CO<sub>2</sub> levels several-fold higher than the atmosphere due to intense root and microbial respiration, with values up to ~4%

https://doi.org/10.5194/egusphere-2025-5242 Preprint. Discussion started: 12 November 2025 © Author(s) 2025. CC BY 4.0 License.

(Drever, 1994). Beyond bulk rhizosphere CO<sub>2</sub> enrichment, fungal hyphae themselves can locally elevate CO<sub>2</sub> concentrations and simultaneously reduce O<sub>2</sub>, creating steep microscale gradients. These self-generated microenvironments may alter surface chemistry, increase mineral roughness, and promote stable colonization, thereby amplifying fungal bio-weathering at the μm scale. While traditional views hold that exudates (e.g., simple carbohydrates) in rhizosphere stimulate microbial growth—enhancing decomposer community nutrient cycling, our findings under controlled conditions are thus consistent with processes expected in natural soils, where abiotic and biotic drivers interact to regulate P release, which provides a new insight into rhizosphere P transformation processes. In addition, our results also consistently showed that both abiotic and biotic weathering processes predominately occurred in the cross section of FAP. Given that approximately ~90% of P is stored in the form of apatite on the earth (Chapin et al., 2011), the preferential weathering of CFAP observed in both abiotic and biotic contexts underscores the need to consider crystal face-specific processes in future studies. Linking microscale fungal–CO<sub>2</sub> interactions with larger-scale geochemical cycling will be essential for more accurate predictions of terrestrial and marine P dynamics.

# 5. Conclusions

Fungal colonization and high CO<sub>2</sub> concentration synergistically promoted FAP weathering. Fungal colonization boosted oxalic acid production and induced P depletion zones. Elevated CO<sub>2</sub> enhanced surface roughness which promoted fungal adhesion and growth, further accelerated bioweathering of FAP. These indicate that the P cycling in soil microenvironments with higher CO<sub>2</sub> concentrations (like in the rhizosphere) may be faster. In addition, the alterations of





















morphology and elemental composition in the mycelial-mineral interfaces indicate that both abiotic and biotic weathering processes preferentially occurred in C<sub>FAP</sub>. Given that approximately 90% of P is stored in the form of apatite on the Earth, future studies should pay more attention to weathering processes at the different faces of FAP crystals, for accurately predicting P cycling in terrestrial and marine ecosystems. **Data Availability Statement** The data that support the findings of this study are openly available in Zenodo digital repository (https://doi.org/10.5281/zenodo.17382598) **Author contributions** MS: Writing-original draft, Data curation, Visualization, Methodology, Investigation, Formal analysis, Conceptualization, Funding acquisition. YC, LT and HW: Data curation, Visualization, Methodology, Investigation, Formal analysis, Conceptualization. XS and ZW: Methodology, Investigation. LM and MX: Methodology. GM, KH, ZbL, ML and GG: Writing-review and editing, Methodology, Investigation, Formal analysis, Conceptualization. ZL: Writing-review and editing, Resources, Methodology, Investigation, Funding acquisition, Formal analysis, Conceptualization. **Competing Interest** The authors declare that they have no known competing financial interests or personal relationships that could have appeared to influence the work reported in this paper. Acknowledgements This study was supported by National Natural Science Foundation of China (42403075), National Key R&D Program of China (2023YFC3707600), the Major Scientific and Technological Project

of CNPC (2023ZZ0203) and China Postdoctoral Science Foundation (2024M753324). 322 323 References 324 325 Bertoloni, G., Bertucco, A., De Cian, V. Parton, T. (2006). A study on the inactivation of micro-organisms 326 and enzymes by high pressure CO<sub>2</sub>. Biotechnology and Bioengineering, 95, 155-160. 327 Bogomolova, E.V., Olkhovaya, E.A., Panina, L.K. and Soukharjevsky, S.M. (2003) Experimental study of influence of rocks and minerals chemical composition and surface structure over the lithobiontic 328 fungi colonies morphology. Mikol Fitopatol 37, 1-13. 329 330 Bonneville, S., Bray, A.W. and Benning, L.G. (2016) Structural Fe(II) Oxidation in Biotite by an 331 Ectomycorrhizal Fungi Drives Mechanical Forcing. Environmental Science and Technology. 50, 5589-5596. 332 Bonneville, S., Smits, M.M., Brown, A., Harrington, J., Leake, J.R., Brydson, R. and Benning, L.G. (2009) 333 Plant-driven fungal weathering: Early stages of mineral alteration at the nanometer scale. Geology 334 335 37, 615-618. Chapin, F.S., Matson, P.A. and Vitousek, P.M. (2011) Changes in the Earth System, in: Chapin, F.S., Matson, 336 P.A., Vitousek, P.M. (Eds.), Principles of Terrestrial Ecosystem Ecology, Second ed. Springer, pp. 337 401-422. 338 Cuisinier, F.J.G., Steuer, P., Senger, B., Voegel, J.C. and Frank, R.M. (1992) Human amelogenesis I: High 339 340 resolution electron microscopy study of ribbon-like crystals. Calcified Tissue International 51, 259-341 268. 342 Drever, J.I. (1994) The effect of land plants on weathering rates of silicate minerals. Geochimica et Cosmochimica Acta 58, 2325-2332. 343 Do Nascimento, J. M., Netto, J. A. F. V., Valadares, R. V., Mendes, G. D., da Silva, I. R., Vergutz, L., & 344 345 Costa, M. D. (2021). Aspergillus niger as a key to unlock fixed phosphorus in highly weathered soils. Soil Biology and Biochemistry, 156, 108190. 346 347 Elliott, J.C. (2002) Calcium phosphate biominerals, in: Kohn, M.J., Rakovan, J., Hughes, J.M. (Eds.),

| 348 | Phosphorus: Geochemical, Geobiological, and Materials Importance, pp. 427–453.                                 |
|-----|----------------------------------------------------------------------------------------------------------------|
| 349 | Elser, J.J. (2012) Phosphorus: a limiting nutrient for humanity? Current opinion in biotechnology 23, 833-     |
| 350 | 838.                                                                                                           |
| 351 | Elser, J.J., Bracken, M.E.S., Cleland, E.E., Gruner, D.S., Harpole, W.S., Hillebrand, H., Ngai, J.T.,          |
| 352 | Seabloom, E.W., Shurin, J.B. and Smith, J.E. (2007) Global analysis of nitrogen and phosphorus                 |
| 353 | limitation of primary producers in freshwater, marine and terrestrial ecosystems. Ecology Letters              |
| 354 | 10, 1135-1142.                                                                                                 |
| 355 | Filippelli, G.M. (2008) The global phosphorus cycle: Past, present, and future. Elements 4, 89-95.             |
| 356 | Gadd, G.M. (2010) Metals, minerals and microbes: geomicrobiology and bioremediation. Microbiology              |
| 357 | 156, 609-643.                                                                                                  |
| 358 | Gadd, G.M. (2017) The geomycology of elemental cycling and transformations in the environment.                 |
| 359 | Microbiology spectrum 5, 1-16.                                                                                 |
| 360 | Giri, B., Almer, J.D., Dong, X.N. and Wang, X.D. (2012) In situ mechanical behavior of mineral crystals        |
| 361 | in human cortical bone under compressive load using synchrotron X-ray scattering techniques.                   |
| 362 | Journal of the Mechanical Behavior of Biomedical Materials 14, 101-112.                                        |
| 363 | Gleeson, D.B., Clipson, N., Melville, K., Gadd, G.M. and McDermott, F.P. (2005) Characterization of            |
| 364 | fungal community structure on a weathered pegmatitic granite. Microbial Ecology 50, 360-368.                   |
| 365 | Hoffland, E., Kuyper, T.W., Wallander, H., Plassard, C., Gorbushina, A.A., Haselwandter, K., Holmstrom,        |
| 366 | S., Landeweert, R., Lundstrom, U.S., Rosling, A., Sen, R., Smits, M.M., van Hees, P.A. and van                 |
| 367 | Breemen, N. (2004) The role of fungi in weathering. Frontiers in Ecology and the Environment 2,                |
| 368 | 258-264.                                                                                                       |
| 369 | Howard RJ, Ferrari MA, Roach DH, and Money NP. 1991. Penetration of hard substrates by a fungus                |
| 370 | employing enormous turgor pressures. Proceedings of the National Academy of Sciences of the                    |
| 371 | United States of America 88: 11281–84.                                                                         |
| 372 | Ichijo, T., Yamashita, Y. and Terashima, T. (1992) Observations on the structural features and characteristics |
| 373 | of biological apatite crystals. 2. Observation on the ultrastructure of human enamel crystals. The             |
| 374 | Bulletin of Tokyo Medical and Dental University 39, 71-80.                                                     |

| 375 | Mendes, G.D., Bahri-Esfahani, J., Csetenyi, L., Hillier, S., George, T.S. and Gadd, G.M. (2021a) Chemical                                           |
|-----|-----------------------------------------------------------------------------------------------------------------------------------------------------|
| 376 | and physical mechanisms of fungal bioweathering of rock phosphate. Geomicrobiology Journal 38,                                                      |
| 377 | 384-394.                                                                                                                                            |
| 378 | Mendes, G.D.O., Dyer, T., Csetenyi, L. and Gadd, G.M. (2021b) Rock phosphate solubilization by abiotic                                              |
| 379 | and fungal - produced oxalic acid: reaction parameters and bioleaching potential. Microbial                                                         |
| 380 | Biotechnology 15, 1189-1202.                                                                                                                        |
| 381 | Meng, L.Z., Chen, Y.H., Tang, L.Y., Sun, X.Q., Huo, H.X., He, Y.X., Huang, Y.N., Shao, Q., Pan, S. and Li,                                          |
| 382 | Z. (2024) Effects of temperature-related changes on charred bone in soil: From P release to                                                         |
| 383 | microbial community. Current Research in Microbial Sciences 6, 100221.                                                                              |
| 384 | Mkhonto, D. and de Leeuw, N.H. (2002) A computer modelling study of the effect of water on the surface                                              |
| 385 | structure and morphology of fluorapatite: introducing a Ca <sub>10</sub> (PO <sub>4</sub> ) <sub>6</sub> F <sub>2</sub> potential model. Journal of |
| 386 | Materials Chemistry 12, 2633-2642.                                                                                                                  |
| 387 | Neal, A.L., Bank, T.L., Hochella, M.F. and Rosso, K.M. (2005) Cell adhesion of Shewanella oneidensis to                                             |
| 388 | iron oxide minerals: Effect of different single crystal faces. Geochemical transactions 6, 77-84.                                                   |
| 389 | Rawat, P., Das, S., Shankhdhar, D. and Shankhdhar, S.C. (2021) Phosphate-solubilizing microorganisms:                                               |
| 390 | mechanism and their role in phosphate solubilization and uptake. Journal of Soil Science and Plant                                                  |
| 391 | Nutrition 21, 49-68.                                                                                                                                |
| 392 | Rogers, J.R., Bennett, P.C. and Choi, W.J. (1998) Feldspars as a source of nutrients for microorganisms.                                            |
| 393 | American Mineralogist 83, 1532-1540.                                                                                                                |
| 394 | Ruiz-Agudo, E. and Putnis, C.V. (2012) Direct observations of mineral fluid reactions using atomic force                                            |
| 395 | microscopy: the specific example of calcite. Mineralogical Magazine 76, 227-253.                                                                    |
| 396 | Schulz, A., Vogt, C. Richnow, H.H. (2012). Effects of high CO <sub>2</sub> concentrations on ecophysiologically                                     |
| 397 | different microorganisms. Environmental Pollution, 169, 27-34.                                                                                      |
| 398 | Smits, M.M., Bonneville, S., Benning, L.G., Banwart, S.A. and Leake, J.R. (2012) Plant-driven weathering                                            |
| 399 | of apatite – the role of an ectomycorrhizal fungus. Geobiology 10, 445-456.                                                                         |
| 400 | Sturm, E.V., Frank-Kamenetskaya, O., Vlasov, D., Zelenskaya, M., Sazanova, K., Rusakov, A. and Kniep,                                               |
| 401 | R. (2015) Crystallization of calcium oxalate hydrates by interaction of calcite marble with fungus                                                  |

| 402 | Aspergillus niger. American Mineralogist 100, 2559–2565.                                                     |
|-----|--------------------------------------------------------------------------------------------------------------|
| 403 | Su, M. (2025). Data for "Influences of CO2 and Fungus-Assisted Bioweathering on Fluoridated Apatite"         |
| 404 | [Data set]. Zenodo. https://doi.org/10.5281/zenodo.17382598.                                                 |
| 405 | Su, M., Mei, J.J., Pan, S., Xu, J.J., Gu, T.T., Li, Q., Fan, X.R. and Li, Z. (2023b) Raman spectroscopy to   |
| 406 | study biomolecules, their structure, and dynamics, in: Saudagar, P., Tripathi, T. (Eds.), Advanced           |
| 407 | Spectroscopic Methods to Study Biomolecular Structure and Dynamics. Elsevier, London, United                 |
| 408 | Kingdom, pp. 173-198.                                                                                        |
| 409 | Su, M., Meng, L.Z., Zhao, L., Tang, Y.K., Qiu, J.J., Tian, D. and Li, Z. (2021) Phosphorus deficiency in     |
| 410 | soils with red color: Insights from the interactions between minerals and microorganisms.                    |
| 411 | Geoderma 404, 115311.                                                                                        |
| 412 | Tyrrell, T., 1999, The relative influences of nitrogen and phosphorus on oceanic primary production. Nature, |
| 413 | 400, 525-531.                                                                                                |
| 414 | Zhang, L., Gadd, G.M. and Li, Z. (2021) Microbial biomodification of clay minerals. Advances in Applied      |
| 415 | Microbiology 114, 111-139.                                                                                   |
| 416 |                                                                                                              |

**Figure 1.** SEM imaging of *A. niger* hyphae on the FAP surface in the CPSF@Soil (a, c) and LPSF@Soil (b, d) treatments. The hyphae promote the formation of erosion channels (yellow arrow) on the FAP cross section (a and c).

420 421

417

Figure 2. SEM imaging of A. niger hyphae and secondary minerals on the FAP surface in the CPSF (a, c,

e) and LPSF (b, d, f) treatments.

425

422

423

Figure 3. Raman spectra of P-O vibration on the FAP surface (a for CPSF and b for LPSF treatments) and normalized atomic percent of Ca and P in FIB-prepared hypha-mineral interface profiles (c for CPSF and d for LPSF treatments) and the corresponding Ca/P ratio in these profiles (e). a1, b1: ~20 μm away from the hypha; a2, b2: ~2 μm away from the hypha. The atomic percentage of Ca and P and Ca/P ratio are related to the points shown along the black dotted lines in Figure c and d. The contents of Ca and P showed obvious changes within 0.32 μm below the hypha-mineral interface.

Figure 4. FAP surface morphology (a, b, c, d), surface step height (e, f, g, h), and surface roughness (i) in the  $C_{FAP}@CO_2$  and  $L_{FAP}@CO_2$ . a and b: SEM images of the  $C_{FAP}$  surface at ambient (a) and elevated (b)  $CO_2$  condition. c and d: SEM images of the  $L_{FAP}$  surface at ambient (c) and elevated (d)  $CO_2$  condition. e-h: Surface step height of the  $C_{FAP}$  (e and f) and  $L_{FAP}$  (g and h) at ambient and elevated  $CO_2$  condition. Images on the right are the atomic force microscope (AFM) height 3D image of FAP surface in the corresponding sites. i: The surface roughness (high sensor) of the FAP based on the AFM analysis. Values were presented as mean±standard error (n = 3, three different sites on one FAP section were analyzed). Asterisk (\*) indicated significant difference (P < 0.05) of FAP surface roughness after etching by

elevated CO<sub>2</sub>, while n.s. indicated no significant difference.

Figure 5. The morphology of the FAP surface and the calcium oxalate formed on it in the CPSF@CO<sub>2</sub> (a, b, c) and LPSF@CO<sub>2</sub> (d, e, f) treatments. a and d: abundant columnar structure with reduced sharpness were observed on the C<sub>FAP</sub> surface (a) and induced additional cracks appeared on the L<sub>FAP</sub> surface (d). b and e: two types of calcium oxalate on the FAP surface. c and f: Calcium oxalate accumulated along the hyphae on the C<sub>FAP</sub> surface (c) and exhibited a scattered aggregation on the L<sub>FAP</sub> surface (f).