# Peer review of "Influences of CO2 and Fungus-Assisted Bioweathering"

_EGUsphere, 2025_

## Referee Comment (RC2)

Review of « Influences of CO2 and Fungus-Assisted Bioweathering on Fluoridated Apatite » by Su et al.

Overall, while the manuscript addresses an interesting topic on the combined effects of $CO_2$ and fungal bioweathering on fluoridated apatite, the study is not fit for publication. It requires quite substantial clarifications and strengthening in many aspects of the discussion – I have strong reservations on the main claim pushed by the authors. The introduction relies heavily on reviews rather than primary studies, this limits precision and depth in describing fungal weathering processes and $CO_2$ effects and leaves a feeling of being a bit swallow. Key mechanisms, such as fungal strategies, hyphal acidification, and microbial community alterations, apatite weathering kinetics need more precise and quantitative context. The methods section is sometimes opaque, lacking sufficient detail on experimental procedures, and analytical parameters. Several claims in the results and discussion are either not supported by the data presented or lack sufficient quantitative evidence, particularly regarding Ca/P alterations. As it stands, I am not supportive of publication in Biogeosciences.

1 – Reference list is too heavy on the reviews/synthesis, more or less 1/3 of the total reference falls in this category – and are repeatedly cited for many different processes (e.g., Rawat et al., 2021 – 4 times ; Hoffland et al., 4 times etc.) In my opinion, this indicates a lack of precision and shows superficiality in the introduction.

Lines 61 : There are different types of fungi. Bonneville et al. use ectomycorrhiza (fungi in symbiosis with tree roots), which is an important distinction to make here, as the demand for P is not the same as in saprophytic fungi.

Lines 62 :  What complementary process ?

Lines 67 : The use of Filipelli et al. (2008) is not appropriate here as this work does not show direct apatite dissolution kinetics, especially in relation to pCO2. At least give some reference that measure apatite dissolution kinetic as a function of pH. There are plenty.

Lines 69 : The same issue with the use of Drever et al., 1994. Again a review – please cite some actual study that measured or calculated pCO2 in rhizosphere or bulk soil instead of relying on reviews.

Lines 71 : The acidification near hyphae at the microscale was actually measured in Bonneville et al. 2011 and again Schmalenberger et al. 2015. Please cite primary studies.

Lines 72 and 73 : In what ways CO2 alters the structure of microbial communities ?

The introduction is not very exhaustive with respect to primary studies (i.e., no reviews!) of fungal bioweathering. Some papers that could be cited include Rosling et al. (2007) in Geobiology or Smits et al. (2014 see below for references), which provide field evidence questioning the efficiency of fungal weathering of apatite. Including such studies would give the introduction more breadth, which currently feels quite shallow. For example, the manuscript discusses the effect of CO2 on apatite weathering, but rates and kinetics—a quantitative aspect—are completely left out. The introduction needs to be more precise and detailed; there are too many shortcomings at the moment for the manuscript to be acceptable for Biogeosciences.

Line 104 : « Sterile water » ? OK, but what type of water ?? MilliQ ? Be precise

Line 103-106 : This protocol is quite opaque to readers who are not familiar with growing fungus. It should be rewritten more clearly.

Line 110 : Appetite ? This must be an error.

Line 110-111 : Please provide the Miller indices of the apatite face exposed to weathering

Line 131 : What is the P content of the PDA medium used in section 2.4 ? Are those P-limited conditions ?

Line 150-151 : I wonder about the effect of applying sterile water at the end of each weathering experiment ? Why is this treatment applied? In my experience with fungal weathering experiments, fungi can fragment rock substrate into very fine particles that are likely lost during this treatment, not to mention the potential dissolution of those colloidal/nanoparticle. The addition of water and its subsequent drying can also be induce the precipitation of secondary phases indepedently of fungal colonization. Was any control trial performed on non-exposed apatite slices to see the effect ? Overall I think this is not a good idea and the potential effect of this treatment should be discussed.

Line 156-158 : For EDS analysis, the count rate (counts·s$^{-1}$) is important—please indicate this. .

Line 166-169 : Concerning TEM-EDS measurements, what is the spot size of the beam, the step size along the profile, and the accelerating voltage used. What X-ray lines were analyzed and importantly the count s-1. There is a lack of explanation on how the Ca and P peak are quantitatively measured from EDS (e.g., background substraction). There is some ambiguity as the measurement are said to be semi-quantitative and yet Fig. 3 presents quantification of Ca and P and of their ratio.

Line 207-208 : How can those numbers be quantified if the TEM_EDS is said to be semi-quantitative (see line 168).

Line 220 : « -9-9 nm » ? Must be a mistake.

Line 238 : what is a « screw dislocation » ? This term must be defined.

Line 243-244 : « *A. niger induced more P depletion zones on the CFAP (Figs. 3a, c), suggesting that CFAP is more vulnerable to fungal weathering than LFAP.* »
This statement is not supported by the data shown in Figure 3. The data do not show a convincing Ca/P increase, synonymous of P depletion. At best, the topmost data point in CFAP show some P depletion but the rest of the Ca/P profile in CFAP (and LFAP) is within the bulk average meaning that there is no P alteration. The constant of Ca and P decrease at depth (in % atomic) are indicative of a thickness effect (i.e., the FIB foil is thinning out toward the top). In TEM-EDS, % atomic percent is not a concentration per unit volume, this is a normalized ratio of detected X-Ray intensities. Thickness variations affect the signals of elements differently. P (and O) emits lower energy X-rays than Ca, so its signal is absorbed more strongly as thickness increase. In addition, O (which contributes a large portion of the total signal) is strongly affected by thickness, which can bias normalized ratios. As a result, the relative percentages of Ca and P can change with depth even when their true concentrations do not.
My advice : work with smaller depth profile – say 500 nm within FAP or even smaller to minimize thickness change effect and detect some feintier chemical alteration. Use

STEM-EDS (instead of TEM-EDS) that focused beam down to a few nm and allow much finer characterization. After all, 45 days of alteration is not much, in order to gain time, you need to look small.

Line 255-257 : « *The results that more calcium oxalate were formed near the mycelium on the CFAP than LFAP (Figs. 2c, d), further confirmed that the bioweathering of CFAP is stronger than that of LFAP* »
This claim is only vaguely supported by the data presented. In figure 2, there is only a small portion of the mycelium network shown (is that representative ?) Using a collection of SEM images, it would have been easy to count Ca-oxalate crystals on the two treatments and make a stronger case for that there is indeed a difference between to two treatment.

Line 257- 263 : « *A. niger can also accelerate the physical destruction of FAP through the biomechanical forces of mycelium growth. Fungal appressoria can produce osmotic pressures of up to 10–20 μN/μm2 during hyphal growth, which would substantially accelerate the physical weathering (Hoffland et al., 2004; Howard et al. 1991). Screw dislocations in crystal cross-sections destabilize CFAP structure, enhancing its susceptibility to biomechanical destruction. Moreover, the released fluorine from FAP did not cause evident toxicity on the mineral surface.* »

Again, this whole paragraph is not supported by the data shown. Do the authors observe appressoria (those are recognizable structure)? To my knowledge, *Aspergillus niger* do not form appressoria. As for the biomechanical forcing, this could have been shown as in Bonneville et al., 2009 (https://doi.org/10.1130/G25699A.1) looking at crystal orientation by SAED using SAED (electron difftraction in TEM), but no such data presented.

Lines 264 -294 : This section is not very convincing. OK there might be a rougher surface developping on apatite crystal due to soil and fungal respiration but this discussion lacks nuance. First, acidification near hypha due to respiration was shown before on a number of rock substrate (see Schmalenberger et al, 2015 -https://doi.org/10.1038/srep12187), then Smits et al. (2014) -https://doi.org/10.1007/s11104-014-2222-6  presented field evidence questionning the acceleration of apatite weathering by fungi, in fact this study showed a retarding effect of fungal colonization on apatite weathering under field conditions.

This section is not very convincing. OK, there might be a rougher surface developing on apatite crystal due to soil and fungal respiration, but this discussion lacks nuance. First, acidification near hyphae due to respiration was shown before on a number of rock substrates (see Schmalenberger et al., 2015). Then Smits et al., 2014 presented field evidence questioning the acceleration of apatite weathering by fungi; in fact, this study showed a retarding effect of fungal colonization under field conditions likely due to complex interactions with soil chemistry, microbial communities, and organic matter.  The manuscript would benefit from acknowledging these contrasting observations, discussing the limitations of laboratory-based microcosm experiments, and providing a more balanced interpretation of how fungal respiration and $CO_2$ may affect apatite weathering in both controlled and field-relevant contexts.

---

## Author Comment (AC1)

**Response to Reviewer's Comments**
**All responses, corrections, and changes have been marked as BLUE color.**

General Comments

Introduction is well structured. More detail should be added however on the selection of the Aspergillus strain. For example, environments found in and function, use in other rock weathering studies, use in industry (strains used for citrate production). This is important justification which grounds the theory present in the introduction to the specifics of this study and the use of Aspergillus.

We have added the details about the *Aspergillus* strain in the Introduction. **Please refer to line 65-70 page 3.**

There are some methodological issues (for example, lack of control treatments and number of replicates) that minimise the power of the analyses and conclusions.

1) The study primarily aimed to QUALITATIVELY (rather than quantitatively) evaluate fungus-Assisted bioweathering on fluoridated apatite. In this study, four experimental pairs were conducted, each comprising two treatments: (i) CPSF@Soil and LPSF@Soil, (ii) CPSF and LPSF, (iii) CFAP@$CO_2$ and LFAP@$CO_2$, and (iv) CPSF@$CO_2$ and LPSF@$CO_2$.

2) For the soil incubation experiments, fungal effects were evaluated by directly comparing apatite surface micro-morphology at hyphae-attached sites with adjacent hyphae-free sites on the same apatite thin section within the same microenvironment. This within-sample comparison effectively minimized soil heterogeneity and environmental variability, allowing the fungal influence to be isolated at the microscale. Therefore, an external abiotic soil control was not required for the primary objective of these experiments.

3) For the culture experiments, pristine apatite and single-factor treatments (fungi or elevated $CO_2$ alone) served as **control treatment** to assess individual and combined effects. Therefore, the absence of a fixed control treatment does not affect the analysis of the results.

4) In addition, as the primary aim of this work is qualitative and mechanistic, focusing on reproducible micro-scale dissolution and alteration features, the study does not include traditional statistical replication. Each apatite thin section was prepared and examined independently at least twice, and the same surface features and dissolution patterns were consistently observed, supporting the robustness of the qualitative conclusions.

The discussion requires significant work to bolster conclusions by referencing to relevant literature, appropriately calling out results and adjusting the structure for readability. Numerous unsupported claims are made that require support from the literature.

We have restructured the Discussion section to improve readability and logical

flow. In addition, we have added relevant references to support key statements.

Specific Comments

Lines 80 -84: two sentences discuss results presented in supplementary information in introduction. Understandable as to why, however results/context from other work should be presented here, discussion of results should be saved for discussion.

We have already adjusted the relevant information into the discussion. **Please refer to line 262-265 page 13.**

Two replicates per treatment were used – the statistical power of this should be noted in the statistics section. More replicates would be preferred or the adoption of a gradient method (e.g. a range of CO2 concentrations).

Statistical analysis was performed only for the assessment of the effects of elevated $CO_2$ on apatite surface roughness. For each treatment (elevated $CO_2$ and ambient atmospheric $CO_2$), atomic force microscopy (AFM) analyses were conducted on three randomly selected surface locations of the apatite samples. The roughness values obtained from these three locations were used for statistical analysis. We have clarified this in the Statistics section of the manuscript. **Please refer to line 199-205 page 10.**

Section 2.3 details the soil experiments. This is difficult to follow, and would benefit from stating the number of bottles used, consistent use of 'samples' or 'glass sheets', and explanation of why all FAP slides were not removed after 30 days (only 2 what about the others?).

To ensure consistent incubation conditions, glass slides carrying cross and longitudinal sections of fluorapatite (FAP) were incubated together within the same glass bottle and considered as two treatments.

After 30 days of incubation, all glass slides carrying FAP sections were removed from the bottles, resulting in a total of four slides (two cross and two longitudinal sections). We have revised Section 2.3 to clearly state the number of bottles, slides, and the terminology used, and to clarify the sampling procedure. **Please refer to line 144-146 page 7.**

A potentially significant issue is the lack of controls, it is not clear that incubations were performed without Aspergillus inoculation in the soil experiments. For example, in the soil incubation the role of abiotic soil factors on weathering was not accounted for (or appear not to be). The soil was acidic, which could result in weathering.

In the soil incubation experiments, treatments without fungal inoculation were not included, as the primary objective was to compare the relative weathering effects of phosphate-solubilizing fungi on different crystallographic orientations of fluorapatite rather than to quantify absolute biotic versus abiotic weathering rates.

Cross and longitudinal fluorapatite sections were incubated together in the same glass bottle, ensuring identical soil chemistry, moisture, pH, and microbial conditions.

After incubation, etch pits were observed exclusively on the transverse sections, whereas no such features were detected on the longitudinal sections. Although the acidic soil may have induced some abiotic apatite weathering, this effect would have acted equally on both crystal orientations under the shared conditions and therefore cannot explain the contrasting surface features.

We note that the acidic soil environment may partly explain why no distinct etch pits were observed on apatite surfaces in subsequent culture-medium experiments following fungal inoculation. However, this does not contradict our main conclusion that phosphate-solubilizing fungus exerts intense weathering effect on the cross-section of fluorapatite. We have added clarification of this experimental rationale and limitation in the revised manuscript.

How were cell concentrations determined? Not presently stated.

1) Spore concentrations were determined using a hemocytometer. More specifically, fungal spores were harvested and suspended in sterile water and thoroughly mixed.
2) Spore concentrations were determined using a hemocytometer under a light microscope. An aliquot of the spore suspension was loaded into the counting chamber, and spores were counted in five representative squares according to standard counting rules.
3) The average number of spores was used to calculate spore concentration, taking into account the chamber volume and dilution factor. All measurements were performed in duplicate, and the mean value was used for subsequent experiments.

The method for measuring spore concentration has been added. **Please refer to line 125-126 page 6.**

Soil sterilisation requires detail. Sterilisation at 121 C in a dry sterilisation mode? Using what instrument (autoclave, oven…)? Heat treatment can alter organics structure; did it affect organic P mobilisation from the soil? Gamma sterilisation would be preferable, why was it not used?

1) We have clarified the sterilization procedure in the revised manuscript. Soil samples were sterilized using a steam autoclave at 121 ºC for 1 h. The soils were placed in glass bottles, and the bottle openings were sealed with sterile breathable sealing film, which allows gas exchange while preventing contamination, thereby ensuring effective sterilization.
2) Heat treatment may alter the structure of soil organic matter. However, the red soil used in this study is characterized by low organic matter and P (total P is 0.19 g/kg). Inorganic P (Fe-P and Al-P) dominating the total P pool. Under these conditions, any heat-induced alteration of organic matter is expected to have a negligible effect on organic P mobilization and does not affect the interpretation of the results. Moreover, all treatments were subjected to the same sterilization procedure, ensuring internal consistency and comparability among treatments.

3) In this study, Gamma sterilization was not used as the sterilization at 121 °C is well accepted for fungal experiments.

The addition of a table of treatments in the methods would be a useful look up tool when reading the discussion.

For clarification, all the treatments were listed in Table 1 below.

**Table 1.** All the treatments in the four experiments

| Experiments | Treatment name |
|---|---|
| Bioweathering of FAP in Soil | CPSF@Soil and LPSF@Soil |
| Weathering of FAP in Medium (i: Weathering of FAP by PSF) | CPSF and LPSF |
| Weathering of FAP in Medium (ii: Weathering of FAP by elevated $CO_2$: Weathering of FAP by PSF and elevated $CO_2$) | $C_{FAP}$@$CO_2$ and $L_{FAP}$@$CO_2$ |
| Weathering of FAP in Medium (iii: Weathering of FAP by PSF and elevated $CO_2$) | CPSF@$CO_2$ and LPSF@$CO_2$ |

Line 176: How was roughness gauged. Not enough to say they were compared. How were they compared? Visual assessment is not sufficient, it is a qualitative method. Number and size of pits/etches/deposits is an example of data that could be used. Covered later, to some extent in section 3.3, but more explicit description on how roughness was quantified in methods is required.

We have clarified the method used to quantify surface roughness in the revised manuscript. Surface roughness of the fluorapatite (FAP) slices was evaluated using atomic force microscopy (AFM), which provides quantitative topographic parameters rather than qualitative or visual assessment. Multiple surface parameters, including step height, surface roughness, and adhesion force can be measured via AFM, as commonly reported in previous studies (De Oliveira et al., 2012; Li et al., 2016; Li et al., 2021).

In this study, Roughness values represent the means of measurements obtained from three randomly selected locations on each FAP slice. The detailed procedure has been added to the Methods section. **Please refer to line 164-166 page 8.**

De Oliveira, R.R.L., Albuquerque, D.A.C., Cruz, T.G.S., Yamaji, F.M. and Leite, F.L. (2012) Measurement of the nanoscale roughness by atomic force microscopy: basic principles and Applications, in: Bellitto, V. (Ed.), Atomic Force Microscopy-Imaging, Measuring and Manipulating Surfaces at the Atomic Scale. InTech, pp. 147-174.

Li, M., Wang, L.J., Zhang, W.J., Putnis, C.V. and Putnis, A. (2016) Direct Observation of Spiral Growth, Particle Attachment, and Morphology Evolution of Hydroxyapatite. Cryst Growth Des 16, 4509-4518.

Li, Z.B., Liu, L.W., Lu, X.C., Zhao, L., Ji, J.F. and Chen, J. (2021) Mineral foraging and etching by the fungus to obtain structurally bound iron. Chem Geol 586.

The discussion begins with reference to supplementary figures - given importance to the discussion they should be included in the main body of the manuscript.

Figure S1 has now been moved from the Supplementary Material to the main body of the manuscript.

Lines 234-242: cause of reductions in structural stability of mineral crystals, with regards to 'screw dislocation', not discussed with reference to LFAP and CFAP.

We supplemented the relevant information in the updated discussion. **Please refer to line 262-277 page 13.**

Lines 238-239 require referencing of the relationship of screw dislocation to reduced stability. Further discussion on how screw dislocations and how they result in reduced stability also needed.

The relationship between screw dislocations and the reduction in structural stability of fluorapatite crystals had been added in the discussion, with appropriate literature references. **Please refer to line 262-277 page 13.**

Paragraph 2 discussion should be split into two at "fungal hyphae can accelerate..." because it deals with different weathering concepts/causes than start of paragraph.

The first part of this paragraph provides evidence that *Aspergillus niger* preferentially weathering the surface of CFAP, while the second part elaborates on the mechanism of apatite weathering by *Aspergillus niger*. This paragraph described a continuous and integrated process of *A. niger*-induced bioweathering of FAP, linking surface colonization, nutrient acquisition, and subsequent chemical and physical weathering mechanisms. Therefore, we consider it appropriate to retain them within a single paragraph to preserve the logical continuity of the discussion.

Line 257: Aspergillus niger's ability to accelerate bioweathering through fungal growth requires referencing, specifically with reference to Aspergillus niger.

1) We have added references to substantiate biomechanical contribution of *Aspergillus niger* to mineral weathering in the revised manuscript.
2) Hoffland et al. (2004) emphasized that filamentous fungi can generate turgor pressure and mechanical forces during hyphal growth, inducing micro-fractures when penetrating mineral fissures or grain boundaries.
3) As a typical filamentous fungus, *A. niger* likely exerts similar localized physical stress on mineral surfaces. In addition, Gadd (2007) noted that *A. niger* hyphae can penetrate mineral cracks and exert minor mechanical pressure, although biochemical dissolution via organic acid secretion remains the dominant process. These references have been added to support our discussion of *A. niger*-induced bioweathering mechanisms.

**Please refer to line 293-296 page 14.**

Gadd, G.M. (2007) Geomycology: biogeochemical transformations of rocks,

minerals, metals and radionuclides by fungi, bioweathering and bioremediation. Mycol Res 111, 3-49.

Line 260: the link between screw dislocations and increased susceptibility to bio weathering is inferred not explicitly tested. It should be stated that it 'may' have resulted in increased susceptibility.

Updated. **Please refer to line 297 page 14.**

Line 262: Potential for fluorine toxicity introduced, but not relevant to the rest of paragraph and not fully discussed (how was it determined, why is it relevant…)

The mention of potential fluorine toxicity was indeed not sufficiently discussed and was not directly relevant to the focus of this paragraph. To improve clarity and maintain the focus of the discussion on $CO_2$- and fungus-induced weathering of fluorapatite, we have removed the statements related to potential fluorine toxicity.

Line 264: reference to literature. How does carbonic acid increase weathering?

Apatite minerals weather congruently (that is, the mineral weathers completely to dissolved products in one step) with dissolved carbon dioxide:

$Ca_5(PO_4)_3OH + 4H_2CO_3 \leftrightarrow 5Ca^{2+} + 3HPO_4^{2-} + 4HCO_3^- + H_2O$ (Filippelli, 2008).

Carbonic acid provides protons which first adsorb onto surface phosphate groups, transforming $PO_4^{3-}$ into $HPO_4^{2-}$ and weakening adjacent Ca–P bonds. The protonation on phosphate surface induces local lattice relaxation and bond cleavage, facilitating the release of $Ca^{2+}$ and $HPO_4^{2-}$ into solution (Dorozhkin, 2012).

We have updated this paragraph and added references. **Please refer to line 298-304 page 14-15.**

Dorozhkin, S.V. (2012) Dissolution mechanism of calcium apatites in acids: A review of literature. World Journal of Methodology 2, 1-17

Filippelli, G.M. (2008) The global phosphorus cycle: Past, present, and future. Elements 4, 89-95.

Lines 269-271: conclusion that "PSF would survive and perform its solubilizing ability under elevated CO2" does not follow from results. It can or may survive. Further, that "that elevated $CO_2$ not only accelerates chemical dissolution but also sustains fungal colonization and activity on mineral surfaces" also doesn't follow from the results. The results do not show fungal colonisation was sustained by elevated CO2 rather were not inhibited. Increases in weathering markers between CO2 and fungi+CO2 treatments necessary to support these conclusions. If they are present, explicitly detail them.

1) *Aspergillus niger* dissolves FAP through oxalic acid secretion, leading to the formation of calcium oxalate as a direct bioweathering product. Although the

amount of oxalic acid released by fungi on the FAP surface could not be quantitatively determined, the presence of calcium oxalate provides clear mineralogical evidence of fungal-mediated dissolution for FAP.

2) Calcium oxalate was observed only on FAP surfaces colonized by *A. niger* (see Figs. 6a, b, d, e), whereas $CO_2$ treatment alone did not produce this feature.

3) Under elevated $CO_2$, the FAP surfaces with fungal hyphae exhibited both $CO_2$-induced morphological alterations (hexagonal pyramids and trench-like structures; see Figs. 6a, d) and fungal weathering signatures (calcium oxalate formation) (see Figs. 6b, e). These demonstrates that elevated $CO_2$ not only accelerates chemical dissolution but also sustains fungal colonization and activity on mineral surfaces.

4) Moreover, there were more calcium oxalate formation in the fungi+$CO_2$ treatments than fungi only treatments (see Figs. 3c, d, e and f) which demonstrates that elevated $CO_2$ further promote the weathering of FAP by fungi.

We have made adjustments in the updated manuscript. In this part, we will discuss the impact of $CO_2$ on the surface of FAP. The synergistic effects of $CO_2$ and fungi will be discussed in detail in the next paragraph. **Please refer to line 311-317 page 15.**

Line 272: statement requires referencing.

The relevant literature has already been cited. (Gorbushina, 2007; Warscheid and Braams, 2000). **Please refer to line 315-317 page 15.**

Gorbushina, A.A. (2007) Life on the rocks. Environ Microbiol 9, 1613-1631.

Warscheid, T. and Braams, J. (2000) Biodeterioration of stone: a review. Int Biodeter Biodegr 46, 343-368.

Line 273: statement requires referencing.

We have updated this statement. **Please refer to line 315-317 page 15.**

Line 276-278: statement requires referencing.

The relevant literature has already been cited (Gadd, 2007; Landeweert et al., 2001). **Please refer to line 323 page 16.**

Gadd, G.M. (2007) Geomycology: biogeochemical transformations of rocks, minerals, metals and radionuclides by fungi, bioweathering and bioremediation. Mycol Res 111, 3-49.

Landeweert, R., Hoffland, E., Finlay, R.D., Kuyper, T.W. and van Breemen, N. (2001) Linking plants to rocks: ectomycorrhizal fungi mobilize nutrients from minerals. Trends Ecol Evol 16, 248-254.

Line 280-281: statement requires referencing.

The relevant literature has already been cited (Fierer et al., 2003; Kuzyakov and Blagodatskaya, 2015). **Please refer to line 330-332 page 16.**

Fierer, N., Allen, A.S., Schimel, J.P. and Holden, P.A. (2003) Controls on microbial CO2 production: a comparison of surface and subsurface soil horizons. Global Change Biol 9, 1322-1332.

Kuzyakov, Y. and Blagodatskaya, E. (2015) Microbial hotspots and hot moments in soil: Concept & review. Soil Biol Biochem 83, 184-199.

Line 283-285: traditional view statement requires referencing.

The relevant literature has already been cited (De Sena et al., 2023). **Please refer to line 325 page 16.**

De Sena, A., Mosdossy, K., Whalen, J.K. and Madramootoo, C.A. (2023) Root exudates and microorganisms. Encyclopedia of Soils in the Environment, 343-356.

Technical Corrections

Lines 69-71: sentence requires reference(s).

We have deleted this statement. **Please refer to line 79-83 page 4.**

Line 185: forming should be formed.

Updated. **Please refer to line 208 page 10.**

Citation: https://doi.org/10.5194/egusphere-2025-5242-RC1

---

## Author Comment (AC2)

**Response to Reviewer's Comments**
**All responses, corrections, and changes have been marked as BLUE color.**

Overall, while the manuscript addresses an interesting topic on the combined effects of $CO_2$ and fungal bioweathering on fluoridated apatite, the study is not fit for publication. It requires quite substantial clarifications and strengthening in many aspects of the discussion – I have strong reservations on the main claim pushed by the authors. The introduction relies heavily on reviews rather than primary studies, this limits precision and depth in describing fungal weathering processes and $CO_2$ effects and leaves a feeling of being a bit swallow. Key mechanisms, such as fungal strategies, hyphal acidification, and microbial community alterations, apatite weathering kinetics need more precise and quantitative context. The methods section is sometimes opaque, lacking sufficient detail on experimental procedures, and analytical parameters. Several claims in the results and discussion are either not supported by the data presented or lack sufficient quantitative evidence, particularly regarding Ca/P alterations. As it stands, I am not supportive of publication in Biogeosciences.

1 – Reference list is too heavy on the reviews/synthesis, more or less 1/3 of the total reference falls in this category – and are repeatedly cited for many different processes (e.g., Rawat et al., 2021 – 4 times ; Hoffland et al., 4 times etc.) In my opinion, this indicates a lack of precision and shows superficiality in the introduction.

1) Rawat et al. (2021) is a representative and comprehensive review in the field of phosphate-solubilizing microorganisms, systematically summarizing the mechanisms of microbial phosphate solubilization from a mechanistic perspective. In contrast, Hoffland et al. (2004) is a landmark review on fungal involvement in mineral weathering, which, from an integrated ecological and geochemical perspective, clearly established that fungi actively regulate mineral dissolution and elemental release through both metabolic activity and physical interactions. These two studies therefore provide essential theoretical and conceptual frameworks that are highly relevant to multiple aspects of the present work.

2) Meanwhile, we have carefully re-evaluated the reference list and removed several non-essential citations to improve clarity and conciseness, while retaining only those references that are directly relevant to the objectives and interpretations of this study.

Lines 61 : There are different types of fungi. Bonneville et al. use ectomycorrhiza (fungi in symbiosis with tree roots), which is an important distinction to make here, as the demand for P is not the same as in saprophytic fungi.

Different fungi, even different microorganisms, may have physiological differences, but they also share the same characteristics. Mycorrhizal fungi and saprophytic fungi may have different requirements for phosphorus due to different driving forces, but they can both promote mineral weathering through similar physical and chemical mechanisms. We have updated this description. **Please refer to line 61-63 page 3.**

Lines 62 : What complementary process ?

The complementary process is the combination of biomechanical and biochemical actions. The former is rooted in the apical extension of the hyphae, and the latter is derived from acidolysis and complexolysis by the excreted low-molecular-weight organic compounds.

Lines 67 : The use of Filipelli et al. (2008) is not appropriate here as this work does not show direct apatite dissolution kinetics, especially in relation to pCO2. At least give some reference that measure apatite dissolution kinetic as a function of pH. There are plenty.

We have added the two below references. **Please refer to line 79 page 4**.

Chaïrat, C., Schott, J., Oelkers, E.H., Lartigue, J.E. and Harouiya, N. (2007) Kinetics and mechanism of natural fluorapatite dissolution at 25 °C and pH from 3 to 12. Geochim Cosmochim Ac 71, 5901-5912.

Harouiya, N., Chaïrat, C., Köhler, S.J., Gout, R. and Oelkers, E.H. (2007) The dissolution kinetics and apparent solubility of natural apatite in closed reactors at temperatures from 5 to 50 °C and pH from 1 to 6. Chem Geol 244, 554-568.

Lines 69 : The same issue with the use of Drever et al., 1994. Again a review – please cite some actual study that measured or calculated pCO2 in rhizosphere or bulk soil instead of relying on reviews.

We have updated the reference.

A in-situ measurement result showed that rhizosphere $pCO_2$ can exceed atmospheric levels by one to two orders of magnitude (Gollany et al., 1993). **Please refer to line 81-83 page 4.**

Gollany, H.T., Schumacher, T.E., Rue, R.R. and Liu, S.Y. (1993) A carbon dioxide microelectrode for in situ $\rho CO2$ measurement. Microchemical Journal, 42-49.

Lines 71 : The acidification near hyphae at the microscale was actually measured in Bonneville et al. 2011 and again Schmalenberger et al. 2015. Please cite primary studies.

We have updated the original description.

Lines 72 and 73 : In what ways CO2 alters the structure of microbial communities ? The introduction is not very exhaustive with respect to primary studies (i.e., no reviews!) of fungal bioweathering. Some papers that could be cited include Rosling et al. (2007) in Geobiology or Smits et al. (2014 see below for references), which provide field evidence questioning the efficiency of fungal weathering of apatite. Including such studies would give the introduction more breadth, which currently feels quite shallow. For example, the manuscript discusses the effect of CO2 on apatite weathering, but rates and kinetics—a quantitative aspect—are completely

left out. The introduction needs to be more precise and detailed; there are too many shortcomings at the moment for the manuscript to be acceptable for Biogeosciences.

The research on fungal weathering apatite has been updated, including the two papers (Rosling et al. 2007; Smits et al. 2014). **Please refer to line 70-76 page 4.**

The effects of CO2 on microorganisms, including biomass, turnover rate and community changes were also rewritten. **Please refer to line 84-94 page 4-5**.

For clarification, $CO_2$ can alter the structure of microbial communities in many aspects, e.g., anaerobic and aerobic microorganisms

Rosling, A., Suttle, K.B., Johansson, E., Van Hees, P.A.W. and Banfield, J.F. (2007) Phosphorous availability influences the dissolution of apatite by soil fungi. Geobiology 5, 265-280.

Smits, M.M., Johansson, L. and Wallander, H. (2014) Soil fungi appear to have a retarding rather than a stimulating role on soil apatite weathering. Plant Soil 385, 217-228.

Line 104 : « Sterile water » ? OK, but what type of water ?? MilliQ ? Be precise

The sterile water refers to ultrapure water that has been sterilized again by high-pressure steam at 121 °C for 20 minutes. **Please refer to line 123 page 6.**

Line 103-106 : This protocol is quite opaque to readers who are not familiar with growing fungus. It should be rewritten more clearly.

We have rewritten this section to provide a more step-by-step description of spore collection, filtration, and quantification, explicitly stating the purpose of each step (e.g., removal of mycelial fragments and determination of spore concentration prior to inoculation).

After the surface of the culture medium was fully covered with black spores, the medium was drenched with sterile ultrapure water and the spores were carefully scraped from the plate surface with a fine artist's brush. The suspension was then filtered through a three-layer sterile cheesecloth to eliminate residual mycelial fragments. The concentration of spores was quantified using a hemocytometer under a light microscope before inoculation. **Please refer to line 121-126 page 6**.

Line 110 : Appetite ? This must be an error.
Corrected

Line 110-111 : Please provide the Miller indices of the apatite face exposed to weathering

The cross section of the apatite corresponds to the basal plane (001). The longitudinal section is parallel to the crystallographic c axis and therefore exposes prismatic faces. However, because the apatite specimens were not oriented single crystals, it is not possible to unambiguously assign a specific Miller index to the prismatic surface (e.g., {100} or {110}). Accordingly, the longitudinal surface is described as a prismatic face parallel to the c axis rather than a specific

crystallographic plane.

Line 131 : What is the P content of the PDA medium used in section 2.4 ? Are those Plimited conditions ?

The phosphorus content of potato dextrose agar (PDA) is not a fixed value, as phosphorus is derived primarily from the potato infusion rather than from added inorganic phosphate. Most of the phosphorus in PDA occurs in organic forms, with only trace amounts of inorganic phosphate. PDA is a standard basal medium for fungal cultivation and is generally not considered phosphorus-limited for fungal growth.

Line 150-151 : I wonder about the effect of applying sterile water at the end of each weathering experiment ? Why is this treatment applied? In my experience with fungal weathering experiments, fungi can fragment rock substrate into very fine particles that are likely lost during this treatment, not to mention the potential dissolution of those colloidal/nanoparticle. The addition of water and its subsequent drying can also be induce the precipitation of secondary phases indepedently of fungal colonization. Was any control trial performed on non-exposed apatite slices to see the effect ? Overall I think this is not a good idea and the potential effect of this treatment should be discussed.

1) The apatite slices were incubated in a vertical position in close contact with the PDA medium. At the end of the experiment, residual solid medium and loosely attached Aspergillus niger spores remained on the apatite surfaces, particularly at the contact interface. A gentle rinsing with sterile water was therefore applied to remove these residues and minimize interference with surface observations.

2) We acknowledge that this step may affect extremely fine weathering products; however, water-induced dissolution of fluorapatite is negligible due to its extremely low solubility (Ksp $\approx 10^{-60}$), releasing only trace amounts of phosphate ($<10^{-6}$ mol P L$^{-1}$) under neutral conditions. Thus, this treatment is unlikely to have altered the observed apatite surface features.

3) Regarding the possible precipitation of secondary phases during wetting and drying, the key observations that abundant calcium oxalate formation were spatially associated with fungal colonization and were not observed on apatite surfaces outside fungal contact zones, indicating a biogenic rather than abiotic origin.

Line 156-158 : For EDS analysis, the count rate (counts·s$^{-1}$) is important—please indicate this. .

During the EDS analysis, the count rate is 900-1200 counts·s$^{-1}$, the acquisition (De Oliveira et al.) time is 60 s.

Line 166-169 : Concerning TEM-EDS measurements, what is the spot size of the beam, the step size along the profile, and the accelerating voltage used. What X-ray lines were analyzed and importantly the count s-1. There is a lack of explanation on how the Ca and P peak are quantitatively measured from EDS (e.g., background substraction). There is some ambiguity as the measurement are said to be semi-quantitative and yet Fig. 3 presents quantification of Ca and P and of their ratio.

1) For the TEM-EDS analyses, the spot size of the beam was 2 nm and the accelerating voltage of 200 kV. Elemental quantification was based on the K-series characteristic X-ray lines of Ca (Ca Kα) and P (P Kα). The count rate during acquisition ranged from approximately 900 to 1200 counts $s^{-1}$, ensuring an adequate signal-to-noise ratio while avoiding detector saturation.

2) Peak intensities were obtained from net peak counts following automatic background subtraction implemented in the EDS software. The analyses are described as semi-quantitative because no external standards were used and matrix effects were not fully corrected at the nanometer scale. Nevertheless, the relative variations in Ca and P intensities and the resulting Ca/P ratios shown in Fig. 3 reliably reflect compositional trends along the profiles rather than absolute concentrations. We have clarified this point in the revised text to avoid ambiguity.

In the revised manuscript, we have added a detailed description of the TEM-EDS analytical conditions and quantification procedure. **Please refer to line 187-192 page 9**.

Line 207-208 : How can those numbers be quantified if the TEM_EDS is said to be semi-quantitative (see line 168).

The analyses are described as semi-quantitatively because no external standards were used and matrix effects were not fully corrected at the nanometer scale. Nevertheless, the relative variations in Ca and P intensities and the resulting Ca/P ratios shown in Fig. 3 reliably reflect compositional trends along the profiles rather than absolute concentrations. We have clarified this point in the revised text to avoid ambiguity.

Line 220 : « -9-9 nm » ? Must be a mistake.
Corrected.

Line 238 : what is a « screw dislocation » ? This term must be defined.

A dislocation is a type of crystallographic line defect representing a one-dimensional imperfection in the crystal lattice, where the regular arrangement of atoms is locally disrupted. Dislocations are characterized by a Burgers vector that describes the magnitude and direction of lattice distortion. A screw dislocation is a specific type of dislocation in which atomic planes are displaced in a helical manner around the dislocation line, with the Burgers vector parallel to the dislocation line.

Such defects can locally increase surface energy and provide preferential pathways for crystal growth or dissolution.

This term had been defined. **Please refer to line 264-268 page 13.**

Line 243-244 : « A. niger induced more P depletion zones on the CFAP (Figs. 3a, c), suggesting that CFAP is more vulnerable to fungal weathering than LFAP. » This statement is not supported by the data shown in Figure 3. The data do not show a convincing Ca/P increase, synonymous of P depletion. At best, the topmost data point in CFAP show some P depletion but the rest of the Ca/P profile in CFAP (and LFAP) is within the bulk average meaning that there is no P alteration. The constant of Ca and P decrease at depth (in % atomic) are indicative of a thickness effect (i.e., the FIB foil is thinning out toward the top). In TEM-EDS, % atomic percent is not a concentration per unit volume, this is a normalized ratio of detected X-Ray intensities. Thickness variations affect the signals of elements differently. P (and O) emits lower energy Xrays than Ca, so its signal is absorbed more strongly as thickness increase. In addition, O (which contributes a large portion of the total signal) is strongly affected by thickness, which can bias normalized ratios. As a result, the relative percentages of Ca and P can change with depth even when their true concentrations do not. My advice : work with smaller depth profile – say 500 nm within FAP or even smaller to minimize thickness change effect and detect some feintier chemical alteration. Use STEM-EDS (instead of TEM-EDS) that focused beam down to a few nm and allow much finer characterization. After all, 45 days of alteration is not much, in order to gain time, you need to look small.

1) We thank the reviewer for the insightful comment and agree that thickness effects may bias TEM-EDS-derived normalized atomic percentages (especially for low-energy X-ray emitters like P and O) and that STEM-EDS-offering more accurate, spatially resolved compositional data-will be a valuable approach.

2) Nevertheless, the present data still provide evidence for localized P depletion at the hypha-mineral interface in the cross-sectional foil. In the CFAP sample, the uppermost data point within ~0.32 μm from the mineral surface consistently shows a decrease in P relative to Ca, whereas deeper portions of the profile converge toward the bulk Ca/P ratio. This observation suggests that the fungal influence on apatite chemistry is spatially restricted to a narrow near-surface zone.

3) For thickness effects, we note that the FIB foils used in this study have an overall thickness of approximately ~70 nm, as confirmed by their high transparency under 4 kV accelerating voltage in the TEM. Within the top ~0.32 μm of the analyzed region, the foil thickness is effectively uniform, and no systematic thinning is observed in this near-surface interval. Therefore, the observed Ca/P variation in this zone cannot be readily attributed to thickness-related absorption effects and is more likely to reflect genuine near-surface chemical modification.

Line 255-257 : « The results that more calcium oxalate were formed near the mycelium on the CFAP than LFAP (Figs. 2c, d), further confirmed that the bioweathering of CFAP is stronger than that of LFAP » This claim is only vaguely supported by the data presented. In figure 2, there is only a small portion of the mycelium network shown (is that representative ?) Using a collection of SEM images, it would have been easy to count Ca-oxalate crystals on the two treatments and make a stronger case for that there is indeed a difference between to two treatment.

1) We agree that the original statement was overstated based on the limited SEM field of view shown in Fig. 2. We have revised the text and weakened the claim.

2) Our interpretation was not based solely on the SEM images, but on a combined assessment of SEM-observed surface grooves and etching features together with AFM data showing more pronounced surface roughening on CFAP. Taken together, these observations suggest enhanced localized dissolution on CFAP.

3) We acknowledge that systematic counting of calcium oxalate crystals from a larger set of SEM images would provide stronger statistical support. While SEM allows reliable identification of micrometer-scale calcium oxalate crystals due to their characteristic morphologies, nanoscale calcium oxalate particles are difficult to fully resolve, meaning SEM-based counts would likely underestimate total calcium oxalate formation. To avoid overinterpretation, we have accordingly toned down the wording in the revised manuscript. **Please refer to line 290-292 page 14.**

Line 257- 263 : « A. niger can also accelerate the physical destruction of FAP through the biomechanical forces of mycelium growth. Fungal appressoria can produce osmotic pressures of up to 10–20 μN/μm2 during hyphal growth, which would substantially accelerate the physical weathering (Hoffland et al., 2004; Howard et al. 1991). Screw dislocations in crystal cross-sections destabilize CFAP structure, enhancing its susceptibility to biomechanical destruction. Moreover, the released fluorine from FAP did not cause evident toxicity on the mineral surface. » Again, this whole paragraph is not supported by the data shown. Do the authors observe appressoria (those are recognizable structure)? To my knowledge, Aspergillus niger do not form appressoria. As for the biomechanical forcing, this could have been shown as in Bonneville et al., 2009 (https://doi.org/10.1130/G25699A.1) looking at crystal orientation by SAED using SAED (electron difftraction in TEM), but no such data presented.

1) We clarify that the *A. niger* strain used in this study does not form appressoria, and no appressorial structures were observed on the apatite surface. Accordingly, we did not intend to claim that *A. niger* generates osmotic pressures of 10-20 μN/μm² through appressoria in our system. The cited values (Howard et al., 1991; Hoffland et al., 2004) were introduced

only to illustrate one possible biomechanical pathway by which fungal growth can contribute to mineral weathering, rather than as direct evidence applicable to our observations.

2) Generally, biomechanical effects associated with fungal growth do not exclusively rely on appressoria. Because fungal hyphal growth is restricted to the tip (Riquelme, 2013), polarized tip extension can generate localized mechanical stresses and tensile forces at the hypha-substrate interface (Howard et al., 1991), which may contribute to physical disruption of mineral surfaces under certain conditions. In the present study, however, we acknowledge that direct evidence for such biomechanical forcing by *A. niger* on fluorapatite is limited.

3) We have updated this part. **Please refer to line 293-296 page 14.**

Riquelme M. (2013) Tip growth in filamentous fungi: A road trip to the apex. Annual Review of Microbiology 67, 587-609.

Lines 264 -294 : This section is not very convincing. OK there might be a rougher surface developping on apatite crystal due to soil and fungal respiration but this discussion lacks nuance. First, acidification near hypha due to respiration was shown before on a number of rock substrate (see Schmalenberger et al, 2015), then Smits et al. (2014) presented field evidence questioning the acceleration of apatite weathering by fungi, in fact this study showed a retarding effect of fungal colonization on apatite weathering under field conditions. This section is not very convincing. OK, there might be a rougher surface developing on apatite crystal due to soil and fungal respiration, but this discussion lacks nuance. First, acidification near hyphae due to respiration was shown before on a number of rock substrates (see Schmalenberger et al., 2015). Then Smits et al., 2014 presented field evidence questioning the acceleration of apatite weathering by fungi; in fact, this study showed a retarding effect of fungal colonization under field conditions likely due to complex interactions with soil chemistry, microbial communities, and organic matter. The manuscript would benefit from acknowledging these contrasting observations, discussing the limitations of laboratory-based microcosm experiments, and providing a more balanced interpretation of how fungal respiration and $CO_2$ may affect apatite weathering in both controlled and field-relevant contexts.

The increased surface roughness of apatite observed in this experiment is caused by high concentrations of $CO_2$ rather than by fungi (see Fig. 5).

We acknowledge the limitations of the laboratory microcosm experiment. The discrepancies between the laboratory results and field observations may arise from the complex interactions with soil chemical properties, microbial communities and organic matter. In particular, the microbial weathering on minerals in field usually points to overall influences by various microorganisms. This study aims to evaluate the weathering effects on different mineral faces by one typical phosphate--sulubilizing fungus. We have provided relevant comparisons and explanations in the Discussion section. **Please refer to line 334-342 page 16.**